# Molecular Mechanisms of High-Altitude Acclimatization

**DOI:** 10.3390/ijms24021698

**Published:** 2023-01-15

**Authors:** Robert T. Mallet, Johannes Burtscher, Vincent Pialoux, Qadar Pasha, Yasmin Ahmad, Grégoire P. Millet, Martin Burtscher

**Affiliations:** 1Department of Physiology and Anatomy, University of North Texas Health Science Center, Fort Worth, TX 76107, USA; 2Department of Biomedical Sciences, University of Lausanne, CH-1005 Lausanne, Switzerland; 3Institute of Sport Sciences, University of Lausanne, CH-1005 Lausanne, Switzerland; 4Inter-University Laboratory of Human Movement Biology EA7424, University Claude Bernard Lyon 1, University of Lyon, FR-69008 Lyon, France; 5Institute of Hypoxia Research, New Delhi 110067, India; 6Defense Institute of Physiology & Allied Sciences (DIPAS), Defense Research & Development Organization(DRDO), New Delhi 110054, India; 7Department of Sport Science, University of Innsbruck, A-6020 Innsbruck, Austria; 8Austrian Society for Alpine and High-Altitude Medicine, A-6020 Innsbruck, Austria

**Keywords:** altitude, hypoxia, acclimatization, oxidative stress, redox homeostasis, mitochondria, genes

## Abstract

High-altitude illnesses (HAIs) result from acute exposure to high altitude/hypoxia. Numerous molecular mechanisms affect appropriate acclimatization to hypobaric and/or normobaric hypoxia and curtail the development of HAIs. The understanding of these mechanisms is essential to optimize hypoxic acclimatization for efficient prophylaxis and treatment of HAIs. This review aims to link outcomes of molecular mechanisms to either adverse effects of acute high-altitude/hypoxia exposure or the developing tolerance with acclimatization. After summarizing systemic physiological responses to acute high-altitude exposure, the associated acclimatization, and the epidemiology and pathophysiology of various HAIs, the article focuses on molecular adjustments and maladjustments during acute exposure and acclimatization to high altitude/hypoxia. Pivotal modifying mechanisms include molecular responses orchestrated by transcription factors, most notably hypoxia inducible factors, and reciprocal effects on mitochondrial functions and REDOX homeostasis. In addition, discussed are genetic factors and the resultant proteomic profiles determining these hypoxia-modifying mechanisms culminating in successful high-altitude acclimatization. Lastly, the article discusses practical considerations related to the molecular aspects of acclimatization and altitude training strategies.

## 1. Introduction

Acute exposure to high altitude imposes hypobaric hypoxia, raising the risk of high-altitude illnesses (HAIs) in inadequately acclimatized individuals [1,2,3]. Such illnesses include acute mountain sickness (AMS), the most frequently observed but usually benign and self-limited form of HAI, and the rare but life-threatening high-altitude cerebral edema (HACE) and high-altitude pulmonary edema (HAPE) [1,2]. Although hypobaria per se may contribute to HAI symptoms [4,5], the central pathogenetic factor is hypoxia [1,6,7,8]. Appropriate acclimatization to hypoxia, whether hypobaric or normobaric, can minimize or even prevent those illnesses [2,9,10].

Although systemic physiological responses to acute hypoxia and systemic physiological changes occurring during acclimatization are well established after decades of extensive research [11,12,13,14,15,16,17,18,19,20,21,22,23,24,25,26,27,28,29], the molecular mechanisms driving these physiological adjustments are not fully understood [30,31,32,33,34,35]. The discovery and characterization of hypoxia-inducible factors (HIFs), key transcription factors regulating gene expression when cellular oxygen availability declines, and other molecular pathways mediating hypoxia responses [36] have paved the way to a better understanding of molecular processes underlying acclimatization to high altitude/hypoxia [37]. For their contributions to deciphering the molecular mechanisms of oxygen sensing in cells and tissues, William G. Kaelin Jr., Sir Peter J. Ratcliffe, and Gregg L. Semenza were awarded the 2019 Nobel Prize in Physiology or Medicine [38]. Nevertheless, our understanding of the complex cell- and tissue-specific signaling and signal integration leading to high-altitude acclimatization on a systemic level is still in its infancy, in particular when considering differences among individuals and additional environmental factors, such as temperature, humidity or radiation. This review discusses molecular processes mediating acclimatization and high-altitude/hypoxia tolerance.

## 2. Systemic Physiological Responses to Acute High-Altitude Exposure and Acclimatization

The most important physiological responses to acute high altitude/hypoxia exposure comprise hyperventilation triggered by the hypoxic ventilatory response (HVR) [39], hemoconcentration due to enhanced diuresis, and an increase in heart rate and cardiac output as a result of sympathetic activation in hypoxia/at altitude [6,40]. In contrast to peripheral vasodilation, acute hypoxia exposure causes pulmonary vasoconstriction (HPV) and elevated pulmonary arterial pressure [17].

Ventilatory acclimatization to hypoxia (VAH), the progressive increase in ventilation during acclimatization, supports recovery of the initially reduced alveolar (PAO_2_) and arterial (PaO_2_) oxygen partial pressures and arterial oxygen saturation (SaO_2_) [12]. Increased ventilation lowers alveolar (PACO_2_) and arterial (PaCO_2_) partial pressures of carbon dioxide, producing respiratory alkalosis and compensatory renal bicarbonate excretion [41]. Paralleling the increased PaO_2_ and SaO_2_, both hemoconcentration and increased cardiac output (at rest and during exercise) help to maintain tissue oxygen delivery in the hypoxic high-altitude environment [42]. However, the acclimatization process, whether occurring at high altitude or in artificial environments (hypobaric or normobaric chambers, tents or face masks with reduced inspired oxygen), encompasses—in addition to ventilatory acclimatization—a broad spectrum of physiological changes, e.g., cardiovascular, cerebrovascular, metabolic, hematological, neurophysiological, and hormonal adjustments [2,10,43,44,45]. Figure 1 depicts the time courses of selected physiological responses to moderate or high altitude/hypoxia.

Alveolar minute ventilation progressively increases over the first 8–10 days at altitude and then plateaus [12,46]. Sympathetic activation at the onset of acute altitude/hypoxia exposure acutely increases heart rate and systemic blood pressure [47,48]. Over the following 10 days at altitude, heart rate tends to decrease [48,49], while elevated systemic blood pressure persists [47]. Increased P_A_O_2_ with acclimatization may produce some dampening of HPV [50,51]. As the initial hemoconcentration subsides, HIF-1 related upregulation of erythropoietin (EPO) sustains the increases in hemoglobin (Hb) concentration and arterial oxygen content (CaO_2_ = Hb x SaO_2_) after the first 1–2 weeks of high-altitude exposure [52]. The increase in muscle capillarity due to vascular endothelial growth factor (VEGF) up-regulation [53] represents another HIF-1-initiated response to altitude/hypoxia exposure.

In summary, although the physiological changes that enable high-altitude acclimatization are well documented, the extent of these responses varies considerably as a function of the severity of hypoxia (altitude) and additional environmental factors, individual vulnerabilities and pre-exposures, and dietary and activity status. The underlying molecular mechanisms of acclimatization and the factors producing large disparities in interindividual susceptibility to HAIs are the topics of this review. Our main interest thereby relates to the acclimatization process of low-altitude dwellers, and we will refer only marginally to high-altitude populations and genetic adaptations at high altitudes.

## 3. High Altitude Illnesses: Epidemiology and Pathophysiology

This section provides a brief overview of the heterogeneous pathophysiologic processes involved in the development of HAIs, which are largely determined by the diverse responses and susceptibility to hypoxia-associated injury among individuals. A comprehensive understanding of the pathophysiology of HAI is essential for the development of new treatment options, for example, to facilitate successful acclimatization by targeting adaptive or maladaptive signaling pathways and molecular processes.

### 3.1. Acute Mountain Sickness and High-Altitude Cerebral Edema

Acute mountain sickness (AMS) frequently develops in high-altitude visitors and typically follows a benign disease course, with symptoms that often resolve over a few days of acclimatization [2,54]. AMS incidence steeply increases with an ascent in altitude or hypoxia intensity (from about 7% at 2200 m to over 50% at 4559 m) in susceptible individuals not acclimatized to high altitude/hypoxia [55,56]. The incidences of severe AMS and HACE are about 24% and 1%, respectively, at 4000 m altitude [57].

Unacclimatized individuals typically develop AMS symptoms within the first 6 to 12 h of acute high-altitude exposure [1,2]. While headache represents the cardinal symptom, subjects most susceptible to AMS typically develop nausea first [58]. In persons suffering from HACE, altered mental status and ataxia are the most prevalent symptoms and are often accompanied by headache, anorexia, nausea, vomiting, and retinal hemorrhages [59,60]. AMS development in unacclimatized persons may be ascribable to acute cellular and/or systemic responses that are delayed and/or insufficient to maintain oxygen delivery to tissues at high altitudes [2,12,39].

Diagnosis of AMS is usually based on the Lake Louise Scoring system (LLS) or less frequently on the abridged (11-item) version (ESQ-C) of the 67-item Environmental Symptoms Questionnaire (ESQ-III) [2]. The original LLS [61] is a self-assessment questionnaire, rating the severity (no discomfort = 0; mild symptoms = 1; moderate symptoms = 2; severe symptoms = 3) of five main criteria, i.e., headache, nausea, dizziness, fatigue, and difficulty sleeping, but has recently been revised by deleting the “difficulty sleeping” criterion [62].

Increased intracranial pressure, brain swelling and edema formation may constitute AMS pathophysiology and explain the symptoms [63,64,65]. Sequential magnetic resonance imaging (MRI) scans during a 22-h exposure to normobaric hypoxia (FiO_2_: 0.12) demonstrated total brain parenchymal expansion, but only the extent of white matter edema (indicating vasogenic edema) correlated with AMS severity [63]. These authors suggested that veno-compression of the small and deep cerebral veins likely contributes to elevated intracranial pressure and brain swelling [63]. Major AMS symptoms, i.e., headache, may result from activation and sensitization of the trigemino-vascular system via mechanical (e.g., elevated cranial and/or intravascular pressure) and/or biochemical (e.g., reactive oxygen species (ROS), nitric oxide (NO), prostaglandins, inflammatory molecules) factors [19,66,67,68].

Why and when severe AMS progresses to HACE remains unclear. Although AMS and HACE are often considered manifestations on a continuum of cerebral HAIs [69], with mild AMS likely progressing to severe AMS and, in rare cases, even to HACE [69], whether HACE really is a severe form of AMS [1] or rather an independent entity remains to be established. The occurrence of white matter edema in HACE indicates dysfunction or disruption of the blood-brain barrier (BBB), which may be caused by ROS-related membrane destabilization and inflammation and/or local HIF and VEGF activation [70].

Pharmacological treatment options for AMS primarily aim to improve oxygen delivery (e.g., hyperoxic breathing or augmenting hyperventilation by the use of acetazolamide), reduce AMS symptoms (e.g., headache) and/or prevent cerebral edema formation by administration of dexamethasone and/or nonsteroidal anti-inflammatory drugs [1,2,71]. However, prevention of AMS in particular, and HAIs in general, through appropriate acclimatization strategies is the most effective countermeasure.

### 3.2. High-Altitude Pulmonary Edema

Both HACE and HAPE represent life-threatening diseases with ~50% mortality when untreated [71]. Like AMS and HACE, the incidence of HAPE primarily depends on the rate of ascent, the absolute altitude attained and the individual’s susceptibility to HAPE. For instance, in mountaineers with unknown HAPE history who ascended over 4 days to an altitude of 4500 m, the HAPE incidence was 0.2%, but the incidence increased to 6% when ascending in only 1–2 days [71], underscoring the importance of acclimatization to prevent HAPE.

HAPE is a non-cardiogenic pulmonary edema caused by pronounced hypoxic pulmonary vasoconstriction and related elevations of pulmonary-artery and capillary pressures, resulting in a noninflammatory and hemorrhagic alveolar capillary leak [72]. Genetic predisposition has been implicated as a cause of the remarkably robust pulmonary vascular responses to hypoxia in some individuals, which may be attributed to insufficient formation and bioavailability of NO likely associated with high ROS levels in hypoxia/at altitude [73,74].

While in high-altitude pulmonary hypertension (HAPH), inflammatory pathways may importantly contribute to the proliferation of pulmonary artery smooth muscle cells and pulmonary hypertension [75], this may not be the case for HAPE. Although inflammation in HAPE could contribute to increased alveolar-capillary permeability, studies in humans rather indicate that inflammation constitutes a secondary response to the pulmonary edema and/or disruption of the alveolar-capillary barrier, not a primary factor in HAPE pathogenesis [72].

Besides rapid descent and oxygen supplementation, administration of the calcium channel blocker nifedipine has been shown to be effective for HAPE prophylaxis and treatment [17,76], but others have questioned nifedipine’s effectiveness [77].

It must be mentioned that apart from hypoxia, several other environmental and/or behavioral factors could contribute to the heightened risk of serious illnesses at high altitudes [78]. Although a comprehensive examination of these factors and their interactions with hypoxia are beyond the scope of this review, the potential contributions of hypothermia and dehydration to HAIs merit discussion. For example, exposure to cold at high altitudes may predispose mountaineers to dehydration due to elevated cold-diuresis and poor access to fluids [79]. These authors identified an association between the level of dehydration and the risk of AMS [79]. Moreover, exposure to cold and altitude may increase the risk of thrombosis and myocardial infarction [80]. Heavy exercise at altitude exacerbates dehydration, raising the risk of rhabdomyolysis and acute liver injury [81], and intensifying arterial hypoxemia, thereby triggering or accelerating the development of AMS [82]. Furthermore, both severe low ambient temperature and intense physical activity are also implicated as predisposing factors for HAPE development, primarily due to the increase in pulmonary artery pressure [83,84]

## 4. Contributions of ROS to the Development of HAIs

### 4.1. Oxidative Stress in Hypoxia

Both acute [85] and long-term [86] hypoxia can increase oxidative stress in humans. High levels of lipid peroxidation, protein oxidation and carbonylation, tyrosine nitration, and DNA oxidation in plasma/serum, erythrocytes or muscle have frequently been reported following exposure to hypoxia [87]. Conversely, acclimatization to high altitude may trigger adaptive responses to oxidative stress in order to restore REDOX homeostasis. The time courses of oxidative stress vs. antioxidative adjustments are still insufficiently understood and likely depend on the hypoxic dose, individual characteristics and other parameters. Vij et al. reported that after 3 months at 4500 m, oxidative stress biomarkers in the circulation were increased by 65% with only marginal changes in antioxidant enzymes [88]. After 13 months, oxidative stress biomarkers fell to pre-exposure levels, likely because of the activation of enzymatic antioxidant defenses. The increased protection of the Tibetan Plateau population against oxidative stress in skeletal muscle compared to lowlanders of the same ethnicity [89] is evidence of long-term adaptive processes to counteract risks of oxidative damage.

### 4.2. Sources of ROS in Hypoxia

Increased ROS production in hypoxia is clearly established. In normoxia, mitochondrial ROS production parallels the overall rate of oxygen utilization. Paradoxically, despite reduced oxygen availability, mitochondria in hypoxia exhibit increased rates of superoxide (·O_2_-) generation by the electron transfer system [90]. In addition, activation of nicotinamide adenine dinucleotide phosphate (NADPH) oxidases and xanthine oxidase, and nitric oxide synthase (NOS) uncoupling, are the most important non-mitochondrial sources of ROS in hypoxia [91].

Hypoxia increases the cellular reductive state, characterized by NADH and FADH_2_ accumulation, when decreased oxygen availability blunts electron flux in the respiratory chain, thereby attenuating mitochondrial membrane potential. Thus, mitochondrial respiration, which ordinarily powers oxidative phosphorylation, instead provokes electron accumulation within the respiratory complexes and subsequent ·O_2_- formation under hypoxic conditions. ·O_2_- is generated mainly at complex III during hypoxia, but complexes I and II also contribute to elevated ROS formation [92].

Acute hypoxia upregulates the mitochondrial ROS-protein kinase C-ε axis that independently activates the NADPH oxidase (Nox) 2 isoform to generate ROS [93]. Interestingly, Nox is upregulated in hypoxic pulmonary arteries, contributing to increased ·O_2_- release [94].

Depletion of *L*-arginine and/or tetrahydrobiopterin (BH_4_) in hypoxic conditions uncouples NOS, transforming the enzyme from NO to ·O_2_- production [95]. In a vicious cycle, ·O_2_- oxidizes BH_4_ to produce dihydrobiopterin (BH_2_), further reducing BH_4_ availability and thereby exacerbating ·O_2_- overproduction by NOS [96]. In addition, hypoxia inactivates endothelial NOS, intensifying this vicious circle in the vasculature [97].

In hypoxia, ATP is increasingly hydrolyzed to ADP and AMP and further degraded to hypoxanthine and xanthine. Concomitantly, hypoxia triggers proteolytic cleavage of xanthine dehydrogenase (XDH) to xanthine oxidase (XO) [98], which then oxidizes hypoxanthine and xanthine generating H_2_O_2_ and ·O_2_- [99]. XDH may be a more important source of ·O_2_^−^ than XO in hypoxia because the intracellular acidification and increased [NADH]/[NAD^+^] ratios favor NADH oxidation by XDH [98].

### 4.3. Role of ROS in Hypoxic Signaling and Acclimatization

Increasing ventilation is a central adaptive mechanism to hypoxia. Although understanding of ROS contributions to acute responses to hypoxia in oxygen-sensing cells is incomplete, it has recently been suggested that ROS generated in mitochondrial complex I during hypoxia could inhibit K^+^ channels and, thus, trigger cell depolarization thereby activating sensory fibers that stimulate the brainstem respiratory center to elicit hyperventilation [100]. In mice with genetically downregulated complex I-activity, decreased ROS production was associated with a blunted tachypnea during acute exposure to 10% O_2_ [101]. In addition, pharmacological inhibition of complex I with rotenone inhibited carotid body secretory responses to hypoxia [102]. Mitochondrial antioxidants attenuated the carotid body chemo-afferent response to hypoxia by almost 50%, associated with a 20% decrease in the hypoxic ventilatory response [103].

### 4.4. Role of ROS on the Pathogenesis of High-Altitude Illnesses

Since hypoxia can induce structural cellular damage exemplified by the modifications of lipids, proteins, and DNA, extensive evidence has implicated oxidative stress in the development of pathologies induced by hypoxia exposure, including AMS [104], chronic mountain sickness (CMS) [105], HAPE [106], HACE [70] and PAH [107]. Irarrázaval et al. reported a correlation between plasma lipid peroxidation and AMS indicators (Lake Louise score and SaO_2_) at high altitudes [108]. In an elegant study in humans, Bailey et al. demonstrated that brain-specific increases in oxidative-nitrosative stress in hypoxia were positively associated with AMS/headache scores independent of blood-brain-barrier impairment [104]. Moreover, augmented systemic oxidative-inflammatory-nitrosative stress was associated with cognitive decline and depression in individuals with CMS [109]. An extensive review of the literature suggests that likely all HAIs are associated with a REDOX imbalance, resulting in increased circulating lipid peroxidation and decreased levels of the antioxidant enzymes superoxide dismutase (SOD) and glutathione peroxidase (GPx) [110].

There is also evidence supporting the direct role of ROS in the pathogenesis of HAIs. NOX4, for example, maybe one of the major sources of ·O_2_- associated with HAPH. Indeed, both transcription and protein content of Nox4 (main Nox subunit expressed in human pulmonary artery smooth muscle cells) are significantly increased in the lungs of patients with PAH compared to healthy subjects [111]. Confirming these findings, Nox4 expression also was increased in a mouse model of hypoxia-induced PAH [112]. Furthermore, the HPV response to high altitude likely results in vascular smooth muscle contraction. Interestingly, the increased hypoxia-induced ROS in lung cells elicited upregulation of the calcium sensor SMIT1 in the endoplasmic reticulum, further contributing to the increase in intracellular calcium and smooth muscle cell contraction [113]. Other studies [114] reported that intermittent hypobaric hypoxia reduces NO bioavailability in lung vasculature, possibly due to NO scavenging by ·O_2_- mainly produced by Nox4. Confirming these findings are *in-vitro* studies in human and rodent pulmonary artery smooth muscle cells showing that hypoxic downregulation of Nox4 and p22phox blunted ROS content and cell proliferation [111]. These findings thus suggest a role for Nox4 in the vascular remodeling associated with PAH. In addition, in mice, the genetic inhibition of the Nox subunit p47phox blunted the pulmonary vasoconstrictor response to hypoxia [87].

Regarding REDOX mechanisms involved in HACE, increased oxidative stress in the brain of rats exposed to severe hypoxia-induced REDOX-sensitive transcription factor NF-κB, which may contribute to cerebrovascular leakage by upregulating the expression of pro-inflammatory cytokines and cell adhesion molecules [115]. Treatment with antioxidants minimizes high-altitude cerebral edema in rats through the PI3K/AKT-Nrf2 pathway and further decreases pro-inflammatory cytokine expression in the brain [116].

Collectively, these studies indicate that, without compensatory activation of antioxidant enzymes, hypoxia-induced ROS overproduction may contribute to HACE, HAPE, HPV and AMS.

### 4.5. Role of ROS-Induced Activation of the HIFs and Nrf2 on Cellular and Systemic Acclimatization

At the molecular level, acclimatization to high altitude hypoxia is affected by the hypoxia-inducible family of transcription factors (i.e., HIFs) and nuclear factor erythroid 2-related factor 2 (Nrf2). HIF-1 and Nrf2 function as surveillance transcription factors that activate genes mediating cellular responses to hypoxia and oxidative stress, respectively. Acting as physiological signaling molecules, ROS generated in hypoxic cells are central activators of both HIF-1 and Nrf2 [117]. Decreased intracellular oxygen concentrations and increased production of ROS activate the translocation of HIF and Nrf2 from the cytosol to the nucleus, where they and their coactivators assemble at promoters of their target genes.

Mitochondria are the major ROS generators in hypoxic cells [118,119]. The decreased availability of oxygen as a terminal electron acceptor causes a backlog of electrons within the respiratory complexes. Substantial evidence implicates respiratory complex III as the major source of HIF-activating ROS [90]. In complex III, the radical ubisemiquinone, an intermediate in the ubiquinone redox cycle [120], donates a single electron to molecular oxygen forming ·O_2_^−^, an oxyradical and precursor of two highly reactive ROS, peroxynitrite (ONOO^−^) and hydroxyl radical (OH). The Rieske iron-sulfur cluster within complex III is a second source of electrons for generating HIF-1-activating ROS [121].

Mitochondrial ROS inactivate succinate dehydrogenase (respiratory complex II), causing a buildup of succinate, a competitive inhibitor of prolyl hydroxylase [122,123]. By suppressing prolyl hydroxylation and proteasomal degradation of the O_2_-regulated α subunit of HIF, succinate may activate molecular adjustments increasing the body’s resistance to oxygen deficiency [124].

NAD(P)H oxidase is a second potential source of HIF-1α-stabilizing ·O_2_^−^. During hypoxia, mitochondrial ROS activate transcription of mRNA encoding Nox, leading to increased translation and activity of the enzyme [125,126]. The resultant ·O_2_^−^ production stabilizes HIF-1α, which in turn activates its repertoire of hypoxia-adaptive genes. Concordantly, HIF-1α content was elevated in cultured cells over-expressing Nox even in normoxia, and HIF-1α accumulation was intensified during hypoxia [125]. Conceivably, hypoxia-induced mitochondrial ROS could activate expression and synthesis of Nox to contribute to its ·O_2_^−^ production to bolster the hypoxic stabilization of HIF-1α.

Hypoxia-induced suppression of HIF-1α prolyl hydroxylation and proteasomal degradation enables HIF-1α transmigration to the nucleus, where it combines with the constitutively expressed HIF-β subunit. The HIF heterodimer and its coactivator, CREB-binding protein [127], assemble at hypoxia-response elements in the promoters of HIF-inducible genes and activate transcription of genes encoding a host of adaptive proteins that collectively enable cells to survive and function in a low-oxygen environment. The products of the HIF gene program include glucose transporters and the complete sequence of glycolytic enzymes supporting anaerobic ATP production [90,128], ferritin to bind free iron and thereby limit iron-catalyzed ROS formation [129,130], VEGF to promote *de novo* capillary growth, and erythropoietin to increase erythrocyte formation and thereby augment the blood’s oxygen-carrying capacity. Collectively, these gene products support acclimatization to hypoxia by realigning cellular energy metabolism with the reduced oxygen supply.

Hypoxia-induced ROS also may activate Nrf2 to drive the expression of a comprehensive program of genes encoding antioxidant and anti-inflammatory enzymes. Under resting conditions, Nrf2 is retained in the cytosol by the Keap1 complex, where the bound Nrf2 undergoes polyubiquitination, targeting Nrf2 for proteasomal degradation. ROS disrupts the disulfide bonds between Nrf2 and Keap1, allowing the transcription factor to translocate to the nucleus, where it assembles with its coactivator Maf at antioxidant response elements in the promoters of Nrf2 target genes. Transcription of these genes enables the synthesis of a host of antioxidant enzymes, including GPx, glutathione reductase, SOD, catalase, the catalytic subunit of the glutathione-synthesizing enzyme glutamate-cysteine ligase, and NAD(P)H:quinone oxidoreductase-1 [131,132]. Nrf2 also activates the expression of genes encoding heme oxygenase-1 and biliverdin reductase, which degrade heme to bilirubin, an anti-inflammatory metabolite [133,134].

Collectively, activation of HIF-1- and Nrf2-driven gene programs affects a powerful, comprehensive response to the bioenergetic and oxidative challenges imposed by hypoxia (Figure 2). However, empirical evidence for the hypoxic induction of these adaptive genes in human subjects is limited. Studies in humans ascending to altitude are essential to determine the contributions of HIF- and Nrf2-driven gene expression to acclimatization to hypobaric hypoxia.

Hypoxia stabilizes HIF-1α by depriving prolyl hydroxylase (PHD) of its substrate oxygen (O_2_) and by causing electron (e-) accumulation within the mitochondrial respiratory complexes, leading to univalent reduction of O_2_ yielding ·O_2_^−^. The latter inactivates respiratory complex II, causing succinate to accumulate and inhibit PHD by competing with the PHD cofactor α-ketoglutarate (α-KG). By increasing demand on the cell’s antioxidant defenses, ·O_2_^−^ and its ROS progeny hydrogen peroxide and peroxynitrite deplete ferrous iron (Fe^2+^), an essential PHD cofactor, further stabilizing HIF-1α to promote its nuclear translocation. In the nucleus, the α and β HIF subunits combine and, in collaboration with the coactivator CREB-binding protein (CBP), activate genes encoding numerous hypoxia-adaptive proteins, including glucose transporters, glycolytic enzymes, vascular endothelial growth factor and erythropoietin. ·O_2_^−^ and other ROS also liberate Nrf2 from Keap1, allowing Nrf2 to transmigrate to the nucleus, where it joins its coactivator Maf to activate the expression of an array of antioxidant and anti-inflammatory enzymes. Collectively, the products of HIF-1- and Nrf2-activated genes increase resistance to hypoxia-imposed oxidative stress, inflammation and ATP depletion.

## 5. How Do Mitochondria Contribute to Successful Acclimatization?

The greatest part of inspired oxygen is utilized by mitochondria for oxidative phosphorylation to produce adenosine 5′-triphosphate (ATP). Among its pronounced effects on mitochondria, hypoxia jeopardizes cellular energy availability, damages mitochondrial components, modulates mitochondrial mass and dynamics and affects the regulation of mitochondrial cell death pathways. Healthy cells respond to these threats by mobilizing cellular stress responses, in which mitochondria are importantly involved, partly through mitochondrial ROS-signaling [135] and crosstalk with HIFs [3,10]. If the stress responses are effective, several beneficial adjustments ensue and can result in improved resilience against hypoxic stress but also may confer additional health benefits and improve performance. Among the main mitochondrial responses to hypoxia are increased efficiency of oxygen utilization, metabolic reprogramming to reduce oxygen dependence, protection from oxidative stress and the regulation of cell survival pathways. Mitochondria play pivotal roles both in the sensing of stress and cellular stress signaling to the nucleus, which in turn is involved in mitochondrial stress adjustments [136].

Hypoxia is further associated with dynamic shape changes and re-localization of mitochondria, as well as with the induction of mitochondrial biogenesis or clearance (mitophagy) mechanisms. However, complex differences depending on the severity of hypoxia, the resilience of the affected cells and the time course (acute versus long-term outcomes) hamper the understanding of mitochondrial responses to hypoxia and may account for partly conflicting reports in the literature.

### 5.1. Oxidative Phosphorylation Efficiency and Metabolic Re-Modelling

Hypoxia causes an acute shift from oxidative energy metabolism to anaerobic glycolysis, up-regulated by sympathoadrenal activation of glycogenolysis and glycolysis [3]. Acclimatization to hypoxia can dampen this glycolytic shift, as shown in mountaineers in whom 3 weeks of acclimatization at an altitude of 4300 m attenuated skeletal muscle glycolysis and associated lactate accumulation as compared to acute altitude exposure [137].

At the molecular level, hypoxia induces several changes in the electron transfer system, many of them orchestrated by HIFs. Conversely, mitochondria are also important regulators of HIF activation via ROS generated at complex III [90], mitochondrial ROS-induced succinate accumulation [123] and regulation of cellular oxygen levels, which control HIFs [3].

Cells may respond to reduced oxygen availability by increasing the efficiency of residual oxygen utilization [138]. A HIF-1-regulated subunit switch in respiratory complex IV (cytochrome c oxidase) [139] may play a pivotal role in this response. In hypoxia, HIF-1 coordinates transcriptional upregulation of the complex IV subunit COX4-2, substituting for COX4-1 subunits that are degraded via HIF-1-mediated upregulation of the mitochondrial protease LON [139]. A HIF-2-regulated atypical subunit-composition of complex IV has further been shown to underlie the exquisite oxygen sensitivity of the carotid bodies, the major mammalian oxygen sensors mediating systemic ventilatory responses in hypoxia [140]. HIF-1 further modulates levels of hypoxia-inducible gene domain family member 1A, which also modulates complex IV efficiency [141].

A complementary mechanism protecting cells from hypoxia-associated energy deficiency is a shift from oxidative metabolism to oxygen-independent, albeit less efficient, glycolysis [142]. AMP-activated protein kinase (AMPK) and mammalian target-of-rapamycin (mTOR) are important factors in this metabolic remodeling [143], and the reduction of oxidative phosphorylation [144], as well as the upregulation of glycolytic enzymes and glucose transport [145] in response to hypoxia, are partially mediated by HIFs. Among the mechanisms promoting this shift of energy metabolism pathways is the inhibition of respiratory complex I via induction of NADH dehydrogenase [ubiquinone] 1 α subcomplex 4-like 2 [146] and via the complex-I assembly inhibitor microRNA miR-210 [147]. HIF-1 activation further reduces pyruvate oxidation via upregulation of pyruvate dehydrogenase kinase 1 (*PDK1*) [148], which negatively regulates pyruvate dehydrogenase.

Moreover, cellular oxygen levels determine various biochemical reactions related to mitochondrial function. For example, while respiratory complex II functions as succinate dehydrogenase in normoxia, in hypoxic conditions, it may function more prominently as fumarate reductase, which is associated with reverse electron flow to fumarate as an electron acceptor [92].

Finally, the improvement of oxygen supply to cells via cell-environmental changes increases mitochondrial oxygen availability in the longer run (see Figure 3). Such adjustments, for example, the oxygen-carrying capacity of the blood (e.g., via increased erythropoiesis) or enhanced vascularization (including higher capillary densities), are largely mediated by HIFs and their target genes, such as erythropoietin or VEGF, as recently reviewed [3].

### 5.2. Mitochondrial Damage; Reactive Oxygen Species and Inflammation

Oxidative phosphorylation is associated with the production of ROS. Although physiological levels of ROS are now recognized as essential signaling molecules [149], excessive ROS causes oxidative stress, which damages biomolecules [150]. Because the electron transfer system is localized to the inner mitochondrial membrane, mitochondrial components are particularly vulnerable to oxidative stress [151].

Hypoxia and subsequent reoxygenation increase ROS production [3]. This effect is somewhat paradoxical since ROS production is expected to positively correlate with the amount of available oxygen. However, impaired electron transfer in the absence of sufficient oxygen boosts ROS production in hypoxia [3]. Adaptive mechanisms, if effective, then lead to a subsequent reduction of ROS, as demonstrated in cell culture, where ROS production increased during acute hypoxia but fell even below control levels during prolonged hypoxia [148,152]. The reduction of oxidative stress is effected in part by HIF-mediated downregulation of mitochondrial respiration and upregulation of mitochondrial antioxidant defenses [153,154] but also by mitochondrial ROS-induced activation of the major coordinator of cellular antioxidant responses, Nrf2 [3,155].

### 5.3. Mitochondrial Shape Changes and Localization

Hypoxic stress and reoxygenation cause changes in mitochondrial morphology. Such stressors impair oxidative phosphorylation leading to reduced mitochondrial fusion and, thus, a shorter mitochondrial phenotype [156]. Severe hypoxia-reoxygenation stress can further provoke the opening of the mitochondrial permeability transition pore and mitochondrial swelling. These events were associated with abnormal fusion events, leading, for example, to the formation of donut-shaped mitochondria, which were more tolerant of metabolic stress and more capable of mitochondrial regeneration [156].

In pulmonary artery endothelial cells, mitochondria were trafficked toward the nucleus upon hypoxic stress. This mitochondrial translocation elicited ROS accumulation in the nucleus that oxidatively modified bases in the VEGF gene’s hypoxia response element in a manner that augmented HIF-1 activation of the gene [157].

### 5.4. Mitochondrial Biogenesis Versus Clearance of Mitochondria

Analysis of muscle biopsies taken from alpinists climbing Mount Everest revealed that, while subsarcolemmal mitochondrial mass remained stable even after 19 days at 5300 m, the further ascent to 6400 m after 66 days at high altitude was associated with loss of mitochondrial mass, particularly in the subsarcolemmal fraction [158]. This finding suggests that mitochondrial mass is particularly susceptible to protracted hypoxia after prolonged exposure to extreme altitude.

Experiments in cultured cells confirm that severe hypoxia or artificial upregulation of HIF-1 is associated with reduced mitochondrial mass. In renal carcinoma cells lacking the HIF-regulator Van-Hippel Lindau protein, HIF-1 decreases respiration in association with down-regulation of a major coordinator of mitochondrial biogenesis, peroxisome proliferator-activated receptor-γ (PPAR-γ) coactivator-1β (PGC-1β) [159]. In mouse embryonic fibroblasts, prolonged hypoxia-induced HIF-1-dependent mitophagy leads to reductions in mitochondrial mass (based on mitochondrial DNA content) [152]. The E3 ubiquitin ligase Parkin [160] and the mitochondrial outer membrane protein FUNDC1 [161] were found to mediate mitophagy and clearance of mitochondria in hypoxic conditions. Parkin- and FUNDC1-mediated mitophagy may be crucial for mitochondrial quality control by promoting clearance of damaged, and thus dangerous, mitochondria [160,161].

In summary, mitochondria are hypoxia sensors and mediators of cellular stress responses; furthermore, hypoxia also exerts direct effects on mitochondrial morphology, abundance and efficiency (see Figure 3). Overall, the cellular response to hypoxia involves a reduction of oxidative metabolism in favor of anaerobic ATP production. Simultaneously, hypoxic cells upregulate protective mechanisms to prevent oxidative stress, mitochondrial and cellular damage and the induction of cell death pathways.

## 6. A Potential Role of Systemic Energy and REDOX Homeostatic Processes Modulated by the STAT3-RXR-Nrf2 Pathway

Decades of biomedical research have achieved important breakthroughs to overcome high-altitude hypoxia, e.g., climbing high and sleep low; supplemental oxygen; Gamow bags and pharmacological interventions (e.g., dexamethasone and acetazolamide). However, the pace of advances in acclimatization strategies has lagged behind the pace of modern life. Current acclimatization schedules are time- and cost-intensive, requiring weeks at altitude, often at remote locations, making them simply untenable for most individuals. Disrupted energy metabolism and increased oxidative stress and inflammatory signaling due to the generation of ROS are the two biggest hindrances to acclimatization at altitude. Besides individual parameters, high altitude acclimatization fundamentally depends on two factors: hypobaric hypoxia intensity dependent on altitude) and duration time spent at high altitudes).

This section considers the effect of intensity, i.e., altitude of hypoxia exposure. Studies on the effects of high-altitude exposure are categorized on the duration of specific altitude exposures [10]. However, the impact of altitude per se on acclimatization has not garnered the same attention. To address the limitations of current pre-acclimatization protocols and to gain insight into the effects of altitude variation, the effects of exposure to three well-recognized altitude zones (high altitude, very high altitude and extreme altitude) in male Sprague Dawley rats were recently investigated [162]. To understand the effects of altitude variation, the rats were exposed to high (3048 m), very high (4572 m), and extreme (7620 m) altitudes for 24 h [162]. Direct exposure to 7620 m caused 100% mortality, while no mortality was observed at 4572 m. To increase survival at extreme altitudes, rats completed a 10-h acclimatization/pre-exposure regimen at 4572 m followed by 1 h normobaric (sea level) exposure and then were abruptly elevated to 7620 m. Remarkably, the acclimatization regimen lowered the mortality rate to zero. Quantitative proteomics and biochemical assays were performed to understand the protective mechanisms of brief exposure to 4572 m. Lungs exposed acutely to 4572 m demonstrated REDOX stress, cytoskeletal instability and energy dysregulation as confirmed by declining contents of REDOX proteins such as Nrf2, peroxiredoxin 6, glutathione peroxidase 3 and thioredoxin, cytoskeletal elements such as vimentin, actin and tubulin, metabolic enzymes such as retinoid X receptor (RXR), malate dehydrogenase, glyceraldehyde 3-phosphate dehydrogenase, molecules such as STAT3 affecting the global proteome, and down-stream effectors such as calpain 2. During 1 h of normobaria after the pre-exposure/rapid-induction regimen, systemic rescue mechanisms restored these elements, effecting powerful protection from 7620 m exposure resulting in 100% survival.

In summary, this method of rapid acclimatization was found to be effective via the modulation of STAT3-RXR-Nrf2 in rats. The identified strategy provided effective acclimatization against altitudes that were associated with high mortality in unacclimatized rats. If this strategy can be translated to humans, the underlying protein networks that regulate/modulate the hypoxic response can be utilized for assessing an individual’s acclimatization status. In the proposed shortened (total 11 h) acclimatization protocol, proteomic data suggest that STAT3, RXR and Nrf2 interplay could have promoted REDOX homeostasis recovery, inactivated inflammation, reduced protein misfolding and restored alveolar integrity and energy homeostasis. This entire process occurred in lung tissue with suitable indicator plasma proteins. The plasma proteins glutathione peroxidase 3, SOD1, hemopexin, catalase, malate dehydrogenase-1, STAT3, thioredoxin reductase-2, RXR, tubulin, Sult 1A1 and monocyte chemoattractant protein-1 were identified as potential markers for high-altitude acclimatization status. Future research is necessary to establish the suitability of these proteins as markers for human high-altitude acclimatization.

A recent study analyzed the plasma proteomes of humans suffering from AMS at 4300 m [163]. In this study, ADAM metallopeptidase domain 15, CD38, cystatin E/M, thrombomodulin and the HIF-1-induced KIT ligand (KITLG) were considered protective against AMS. Upregulation of phosphoenolpyruvate carboxykinase 1, phosphoglycerate dehydrogenase, ribokinase, S100A12, solute carrier family 4 member 1 and secreted protein acidic and cysteine-rich was suggested to predict AMS. Ten proteins were found to be potentially useful diagnostic markers of AMS: ADP ribosylation factor 6, Epstein-Barr virus induced 3, GC vitamin D binding protein, immunoglobulin superfamily containing leucine-rich repeat 2, myocilin, neuropilin 2, ribokinase, ret proto-oncogene, TNF receptor-associated factor 2, and WAP, follistatin/kazal, immunoglobulin, kunitz and netrin domain containing 1. Although the biological functions (including REDOX regulation, inflammation and vascular function) of some of the acclimatization-associated proteins were identified in the human AMS study [163] and the acclimatization study in rats [162], the translational utility of the proposed acclimatization biomarkers remains to be confirmed.

Aside from and related to changes in acclimatization proteins, genetic differences likely determine high altitude acclimatization and individual vulnerabilities to HAIs. The next section discusses these genetic factors.

## 7. Genetic Aspects of High-Altitude Acclimatization

The mechanistic underpinnings of HAIs are complex, and the multiple clinical symptoms are associated with numerous biological pathways [3,164,165,166,167]. Genetic, transcriptional, translational and metabolic markers of these pathways are thought to link an individual’s biological state with phenotypic responses, thereby shaping the adjustment or maladjustment to the hypobaric hypoxic environment. Over the last decade, extensive investigations of genetic distributions have been performed using candidate gene approaches and advanced techniques such as Next-Generation Sequencing and Genome-Wide Association Studies (GWAS).

Table 1 lists polymorphisms of several of the most important genes (and respective proteins) involved in high-altitude acclimatization. Most of these genes are associated with multiple, frequently interrelated pathways [3,164,167]. Initial investigations mainly focused on genes and proteins impacting the vascular system or endothelial dysfunction, and processes such as the renin-angiotensin-aldosterone system, apelin signaling, and NO signaling were of particular interest. During the last decade, specific hypoxia-induced signaling, particularly that mediated by the HIF family, became a central focus of mechanistic high-altitude research. Genes of the greatest significance for the field include *ACE*, *AGT*, *ADRB2*, *EDN1*, *VEGFA*, *SLC6A4*, *NOS3*, *CYBA*, *PPARA*, *NFE2L2*, *CDIP1*, and *ANGPTL4, HIF1α*, *EGLN1, HIF1AN*, *EPAS1*, and *HMOX2*. Even single-nucleotide polymorphisms in these genes, which play crucial roles in related physiological functions, modulate the efficiency of individual acclimatization (Table 1).

The significant genotypic differences among the world’s high-altitude populations and visiting lowlanders are especially consequential for individuals genetically predisposed to HAI. For example, increased frequency of the G (vs. T) allele of the *NOS3* Glu298Asp (rs1799983G/T) polymorphism is associated with higher levels of the potent vasodilator NO in high-altitude natives or acclimatized lowlanders [168,169,170,171]. Likewise, the G allele of *ADRB2* Arg16Gly (rs1042713A/G) is prevalent in high-altitude natives and acclimatized sojourners, and the SNP has been reported to influence oxygen regulation [172,173]. The oxidative stress-related genes *CYBA* (cytochrome b−245 α polypeptide) and *GSTP1* (glutathione transferase Pi 1) also are potential candidate genes affecting high-altitude acclimatization. *CYBA* (cytochrome b−245 α polypeptide) encodes the p22phox subunit, a critical component of the NADPH oxidase system; *GSTP1* plays a vital role in regulating lipid peroxidation. The alleles A of −930A/G (rs9932581), T of rs4673C/T (H72Y) in *CYBA* and A of I105V (A/G) in *GSTP1* in sojourn controls and healthy high-altitude natives were associated with lower circulating levels of the oxidative stress marker 8-iso-prostaglandin F_2α_ [174]. The physiological relevance of these allelic presentations points to an advantage in maintaining endothelial homeostasis.

Among the genes that contribute to oxygen signaling, the association of *HIF1α* with altitude physiology is surprisingly enigmatic. Two SNPs, rs11549465C/T and rs11549467A/G, are associated with changes in endurance (Table 1). However, the distribution of *EPAS1* and *EGLN1* SNPs and haplotypes have been studied extensively [174,175,176,177,178]. A GWAS conducted by Beall et al. identified an over-representation of eight SNP variants of *EPAS1*, rs1868092, rs1447563, rs11125075, rs4953388, rs4953396, rs896210, rs896210 and rs6735530, that are associated with lower hemoglobin concentrations in Tibetan high-altitude populations [175].

Another GWAS demonstrated an over-representation of several SNPs of *EGLN1* but cited only the two most abundant SNPs, rs1769792 and rs12030600, in Andeans and Tibetans, respectively [177]. Thus, the two populations represent different genotypes in the same gene. *EGLN1* is pivotal to acclimatization and adaptation [174,178]. Mishra et al. investigated seven SNPs of *EGLN1*, rs1538664, rs479200, rs2486729, rs2790879, rs480902, rs2486736 and rs973252, and found them crucial for adaptation and maladaptation in the highland (3500 m) Ladakh population [178]. The haplotype G-C-G-T-T-G-A was over-represented and correlated with elevated SaO_2_ [178]. Deviating from the intronic genotypes of *EGLN1*, in Tibetans, two over-represented exon-1 *EGLN1* variants, Asp4Glu and Cys127Ser, associated with a lower affinity for oxygen, suggesting a gain of PHD2 function and loss of HIF function, but in the presence of co-chaperone p23, a reversal of the function was reported [179,180]. A non-synonymous *EGLN1* SNP rs186996510 (D4E) was found to be significantly prevalent in Tibetans but negligible in Han Chinese, Japanese, Europeans and Africans [181]. Moreover, rs186996510 was associated with higher blood hemoglobin content in Tibetans vs. acclimatized lowlanders at high altitudes. Several reports appeared on the adaptation and maladaptation of these two genes in various native highland populations, especially the Tibetans versus Han Chinese.

The genetic distribution of SNPs may impact physiological acclimatization to hypoxia (Figure 4). In combination, these genotypes may synergistically increase physical performance and enhance endurance. In contrast, these genotypes may intensify discomfort and predispose to various illnesses, both in sojourners and the native highland population.

Complex physiological functions are compound outcomes of several pathways. Here two classes of major pathways, the oxygen-sensing (HIF) and vascular pathways are schematically depicted. Which pathway is the first to influence the other remains debatable. Under hypoxia, the HIF system is instantly active, and the master regulator HIF-1α activates several genes. Here, the genetic variants, specifically in *EGLN1* and *EPAS1*, play a distinct role in impeding HIF-1α function and the redox potential, thus significantly reducing the oxygen and nitric oxide levels. Likewise, the variants in the genes of the vascular system, such as *ACE*, *CYP11B2* and *NOS3*, play a significant role in altering circulating angiotensin-II (AT-II) activities, which powerfully impacts sodium and water retention to elevate blood pressure in the pulmonary system. The synchronization between the two systems ensures low oxygen and NO and high AT-II, affecting HA illnesses. ACE, angiotensin-converting enzyme 1; ADMA, Asymmetric dimethylarginine; ADRB2, adrenoreceptor β-2; AGT, angiotensinogen; ALD, aldosterone synthase; APLN, apelin; EGLN1, Egl nine homolog 1/prolyl hydroxylase domain-containing protein 2; EPAS1, Endothelial PAS Domain Protein 1; EPO, Erythropoietin; HIF-1α, hypoxia inducible factor 1 α; I/D, insertion/deletion; NO, nitric oxide; NOS3, endothelial nitric oxide synthase; VEGF, vascular endothelial growth factor; ↑, elevated level/expression ↓, decreased level/expression.

The role of oxidative stress and vascular markers in AMS is generally important, but here, we focus only on a subset of related genes. *EPAS1* rs6756667 is associated with mild gastrointestinal symptoms and *VEGFA* rs3025039 with a mild headache (Table 1). Accordingly, the interaction of genotypes rs6756667 GG and rs3025039 CT/TT increased the risk of developing AMS [182]. In the SNP rs4953348A/G of the same gene, the G allele frequency correlated with lower levels of SaO_2_ in AMS [183]. Another investigation found that *EPAS1* rs6756667 increased the risk of AMS [184]. Moreover, *EGLN1* 5′-UTR SNPs rs12406290 and rs2153364 were associated with AMS, but the haplotype increased the risk more than the individual SNPs [185].

The *NOS3* SNPs have been extensively investigated in HAPE ([2] and references therein). In Indian sojourners, Japanese and Chinese populations, the T (Asp) allele-associated heterozygous and homozygous genotypes of *NOS3* SNP Glu298Asp (rs1799983T/G) were prevalent in HAPE patients. In the Indian HAPE cases, SaO_2_ and NO levels were reduced in carriers of this allele [168,169]. The oxidative stress candidate markers *CYBA* contributed significantly, the variant alleles G of −930A/G and C of H72Y (C/T) and *GSTP1* contributed G of I105V (A/G), associated with elevated circulating 8-iso-prostaglandin F_2α_ in HAPE [186]. The *EGLN* variants were strongly associated with maladjustment [178].

The established genetic signatures of maladjustment to high altitude are limited to only a few selected genes, with the role of thousands of closely interconnected genes still enigmatic. Like physiological functions in general, acclimatization to hypoxia is multi-factorial and multigenic. The genomic SNPs interact differently, leading to different physiological outcomes (Figure 4). Different significant SNPs in the same gene were identified in diverse populations, possibly due to ethnic and regional variations [165,187,188,189]. Since SNPs are strongly associated with divergence in HAPE severity, proteomic and epigenomic factors could profoundly affect HAPE susceptibility [3,163]. Hence it is expected that multiple SNPs will impact individual vulnerabilities to HAPE. Comprehensive assessment of each SNP’s impact is becoming increasingly urgent.

Genetic predisposition, environmental conditions (primarily hypoxia) and the resulting physiological or pathological responses to high altitude determine the efficiency of acclimatization and the risk of HAIs. Although therefore, these responses are individual, the time course of normal acclimatization to high altitude can be summarized, and deviations may inform about pathological processes related to HAIs. These time courses are discussed in the following section.

## 8. Time Course of Acclimatization and Memory Effects after De-Acclimatization

The time courses of several physiological variables are presented in Section 2 and Figure 1. It becomes evident that short-term acclimatization, in particular, ventilatory acclimatization is almost complete after 10 days at high altitude, whereas more long-term effects, e.g., effective erythropoiesis, require weeks to months [10,12,42,46,190,191,192].

The persistence of hypoxic effects after high-altitude acclimatization, i.e., hypoxic “memory”, is supported by more rapid acclimatization and development of less severe AMS symptoms in human subjects re-exposed to high altitude after a 1–3-week sea level or low-altitude sojourn. For example, Lyons et al. reported significantly reduced AMS-C scores (0.1 vs. 0.6) after re-exposure of 6 male lowlanders to high altitude (4300 m; hypobaric chamber) after 8 days at sea level following initial acclimatization to high altitude (4300 m; real altitude) [193]. Beneficial effects of previous acclimatization were related to increased values of SpO_2_, hemoglobin and hematocrit [193]. Moreover, Beidleman et al. demonstrated an 88% AMS incidence in sea level residents acutely exposed to 4300 m, but after 12 days at altitude and another 12 days at sea level, only 17% of the subjects developed AMS upon acute re-exposure to 4300 m in a hypobaric chamber [194].

A variety of molecular mechanisms may contribute to these observations, e.g., some persistence of HIF-1α regulated VAH [195] and higher plasma adenosine concentrations during the second hypoxia exposure as a result of erythrocyte hypoxic memory [196].

Adjustments of carotid bodies are essential for VAH as the denervation of carotid bodies ablates the phenomenon [197]. Hypoxia sensitivity of glomus cells increases during prolonged (chronic) altitude/hypoxia exposure, associated with changes in chemosensory mechanisms and neurotransmitters but also with structural changes (hyperplasia, angiogenesis) of carotid bodies [195,198]. Both the acclimatization and de-acclimatization time courses depend on the duration and severity of the hypoxia exposure [195]. Of particular note, the presence of HIF-1α is not needed for the normal HVR (at least in mice) but is necessary for VAH and oxygen sensing in the carotid body [199]. Compared to the VAH, the HIF-1-dependent erythropoietic response is considerably delayed. Circulating hemoglobin mass increases very gradually, at a rate of about 1% per 100 h high-altitude/hypoxia exposure [52]. As is true for VAH, this response is essentially independent of the type of hypoxia applied [52,195]. After peaking at about 48 h hypoxia, plasma EPO activity subsides, paralleling a decrease of the soluble form of the EPO receptor, sEPO-R, and then remains slightly elevated, sufficient for continuous induction of erythropoiesis [200]. The small increase in circulating erythrocytes persists until their degradation (3 months half-life). Other mechanisms, such as the upregulation of membrane EPO-R and reduction of sEPO-R, and antiapoptotic factors in erythroid progenitors of the bone marrow, also may contribute to the elevated erythropoiesis in chronic hypoxia [201]. Increased erythropoiesis slackens after about 3 months at high altitudes and stabilizes after about 8 months [202]. Hemoglobin mass, which was increased after 3-week acclimatization at a high altitude (3450 m), returned to baseline within 11 days after returning to a low altitude [203]. The authors suggested that this decrease was not due to neocytolysis but rather to a reduced rate of erythropoiesis accompanied by normal clearance of senescent erythrocytes [203].

## 9. Practical Applications of Acclimatization and Altitude Training Strategies

How can the above-described time courses in physiological responses to hypoxia translate into strategies to help athletes and mountaineers prepare for and/or adapt during altitude exposure? In this section, we discuss the potential influence of 8 prominent acclimatization mechanisms on athletes’ and mountaineers’ preparation for altitude exposure.

### 9.1. Consequences of Ventilatory Acclimatization

Before traveling to a high altitude, several pre-acclimatization strategies, either using intermittent hypobaric [204,205,206,207,208] or normobaric [209,210] hypoxia exposures, proved effective in eliciting ventilatory acclimatization (e.g., decrease in end-tidal PCO_2_ due to hypoxia-induced hyperventilation and increase in SpO_2_). However, when assessed at PO_2_ values equaling those present at an altitude of 4300 m, normobaric hypoxia strategies induced less ventilatory acclimatization than hypobaric hypoxia strategies and did not improve exercise performance [54], suggesting less efficient transfer of the benefits induced by normobaric hypoxia vs. acclimatization to the actual altitude [211]. This discrepancy may explain why elite mountaineers combine normobaric hypoxia pre-acclimatization (e.g., hypoxic tent) with a final preparation in “real” altitude in the Alps before traveling to the Himalayas [212]. Such a pre-acclimatization strategy is effective, imparting ventilatory benefits that last for several days [208,213].

### 9.2. Consequences of Increased Diuresis

The combination of the increase in hyperventilation-induced water loss and the increase in urination for renal bicarbonate excretion [41] to compensate for the respiratory alkalosis leads to higher fluid losses, particularly during the early phase (i.e., acclimatization period) of the altitude exposure [214]. Altitude may also reduce thirst and appetite. These phenomena are challenges to maintaining hydration status [215,216] and may cause severe dehydration in athletes [217,218]. Focus on the hydration status (e.g., monitoring urine specific gravity and/or daily body mass changes) is therefore recommended for lowlanders sojourning at altitude.

### 9.3. Consequences of Shifts in Substrate Oxidation

Altitude exposure modifies substrate oxidation for a given exercise intensity level and lowers the reliance on lipids for a larger while increasing dependence on carbohydrate (CHO) oxidation [219,220]. Shifts toward greater carbohydrate (CHO) utilization at high altitudes [221] have nutritional consequences, including greater dietary CHO requirements to replace muscle glycogen and greater CHO intake during exercise [215,222]. Moreover, since altitude increases leptin levels and, therefore, may reduce appetite [223], monitoring energy intake to limit losses of body mass is paramount in athletes and mountaineers.

### 9.4. Consequences of Sympatho-Adrenal Activation

Sympathetic activation, a pivotal response to altitude [224,225], has important pathophysiological consequences, which were described in Section 2 and Section 3. In elite athletes, to detect over-reaching, the sympatho-vagal balance is indirectly and non-invasively monitored by assessing heart rate variability (HRV) [226,227]. In altitude training camps, the addition of the hypoxic stimulus modifies the HRV parameters, mainly a decrease in RMSSD (root mean square of the successive differences between adjacent RR intervals) and high frequency (HF) variability as well as increases in low frequency (LF) or LF/HF, reflecting either a decrease in parasympathetic or an increase in sympathetic tone branch [228,229]. This autonomic shift necessitates modification of the training content—mainly by decreasing the training intensity—[230] in order to maintain homeostasis and limit the risk of non-functional overreaching.

### 9.5. Consequences of the Shift from Oxidative Energy Metabolism to Glycolysis

Ascent to altitude prompts an acute shift to glycolysis. The enhancement in glycolytic flux might be expected to increase maximal serum lactate, but peak lactate is lowered in altitude, the so-called (and debated) lactate paradox [231,232]. The lactate paradox may arise from a decrease in maximum substrate flux due to the reduced energy supply in hypoxia and/or alterations in the metabolic control of glycogenolysis and glycolysis at the cellular level, due in part to sympatho-adrenal activation [233]. Due to renal bicarbonate excretion and associated reductions in blood buffering capacity, it is generally recommended to avoid exercise with a large glycolytic contribution (e.g., lactate production) during the acclimatization period [234].

### 9.6. Consequences of Hypoxia-Activated Erythropoiesis

The increases in hemoglobin mass and subsequent enhancement in oxygen transport capacity are among the most expected consequences of prolonged exposure to altitude in elite athletes [191]. One important consequence is that the benefits of an altitude camp require measuring the changes in hemoglobin mass (and not hemoglobin concentration, which may be altered by hemoconcentration) by the CO rebreathing method [235]. The altitude-induced hematological benefits are reported even in elite endurance athletes with elevated values of hemoglobin mass and VO_2max_ [236,237], but there is large variability between subjects [234] as well as between exposures for a given athlete [238]. Optimal increase in hemoglobin mass requires appropriate monitoring of the athletes since there are many potential limitations (e.g., energy intake, hydration and inflammatory status) [239], with pre-ascent iron stores (e.g., ferritin level) of the athlete being of particular importance [240]. The optimal amount of iron supplementation in athletes with clinically normal iron stores remains a subject of debate [241].

### 9.7. Consequences of Energy Metabolism Optimization

Athletes training at altitude expect first an enhancement in maximal aerobic power (VO_2_max) after returning to sea level. The increased efficiency of O_2_ utilization at the mitochondrial level is often neglected and may lead to improved economy (e.g., lower O_2_ consumption at a given speed), which is paramount in endurance sports [242]. This increase in economy is largely reported following altitude exposure [243,244] and is likely as important as improved VO_2max_ for post-altitude enhancement in endurance performance at sea level.

### 9.8. Consequences of Oxidative Stress

Oxidative stress may have practical consequences for athletes during hypoxia exposure since it modulates the HVR that is paramount for altitude acclimatization (see Section 9.1) [86].

Antioxidant supplementation—at least during the early phase of altitude camp—may be counterproductive, as it could potentially blunt or delay acclimatization to altitude in athletes [86]. Conversely, it was reported that increases in antioxidant-rich foods intake during altitude exposure produced the expected increase in antioxidant capacity and attenuated some altitude-induced systemic inflammatory biomarkers in elite athletes [239]. Despite a general consensus that the pro-antioxidant balance is paramount for altitude acclimatization, the question of the appropriate nutritional strategy remains unresolved.

## 10. Conclusions

Although our knowledge of the molecular underpinnings of altitude acclimatization has substantially increased in recent decades, the determinants of successful acclimatization to hypoxia and the etiology and risk factors of HAIs remain poorly understood. This review summarized the important contributions of hypoxia-stimulated cellular stress responses, particularly related to cellular REDOX regulation, transcriptional orchestration of hypoxia adjustments and mitochondrial changes. Modulated by individual genetic predispositions, these processes can lead to acclimatization and hypoxia tolerance via favorable metabolic reprogramming, structural and functional mitochondrial responses, the induction of anti-oxidative and anti-inflammatory defenses and the regulation of local and temporal patterns of systemic adjustments.

There is now ample evidence that impairments or attenuation of these processes contribute to the development of all forms of HAIs, including AMS, HACE and HAPE. The activation of gene programs driven by HIF-1, Nrf2 and other regulators of transcription affects a powerful, comprehensive response to the bioenergetic and oxidative challenges imposed by hypoxia. The modifying changes in the proteomic makeup of cells, tissues or circulating media may represent powerful biomarkers for the acclimatization status and diagnosis or prediction of HAIs.

Further characterization is necessary to obtain a deeper understanding of these molecular mechanisms that may help to optimize or design novel prophylactic and therapeutic strategies during ascent to high altitude or exposure to normobaric hypoxia. Consideration of these mechanisms is essential for efficient acclimatization and preservation of performance of commuters, pilgrims, tourists, mountaineers and athletes. Furthermore, mounting evidence suggests that the intentional induction of hypoxia-induced adjustments may be harnessed as a new therapeutic modality, e.g., for the management and treatment of cardiovascular [245], neurological [246] or psychiatric diseases [247].

## Figures and Tables

**Figure 1 ijms-24-01698-f001:**
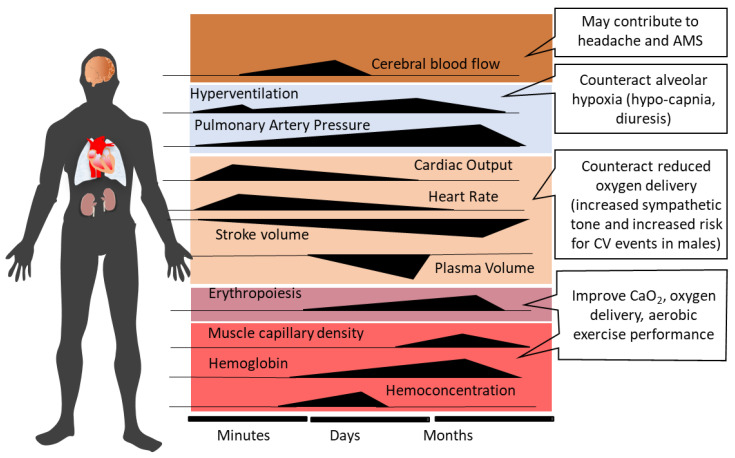
Time-dependent changes and related consequences of physiological responses during acclimatization to high altitude (modified from Burtscher et al., 2022 [10]). AMS, acute mountain sickness; CV, cardiovascular; CaO_2_, arterial oxygen content.

**Figure 2 ijms-24-01698-f002:**
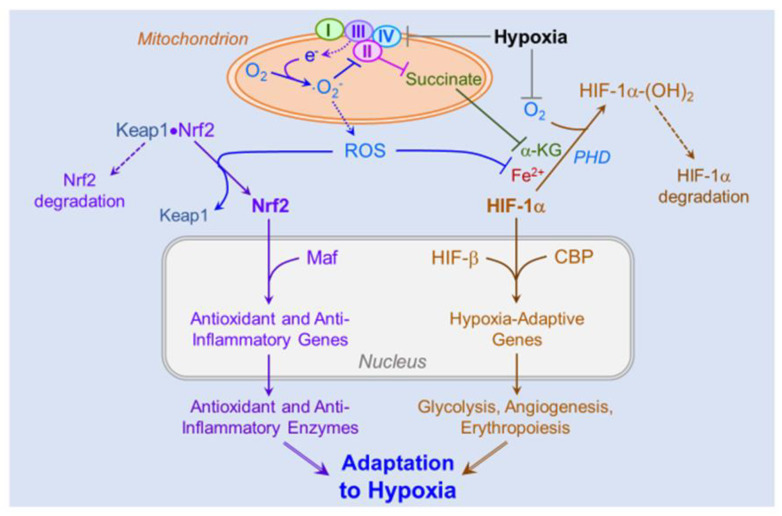
Parallel activation of Nrf2- and HIF-1-responsive gene programs by hypoxia.

**Figure 3 ijms-24-01698-f003:**
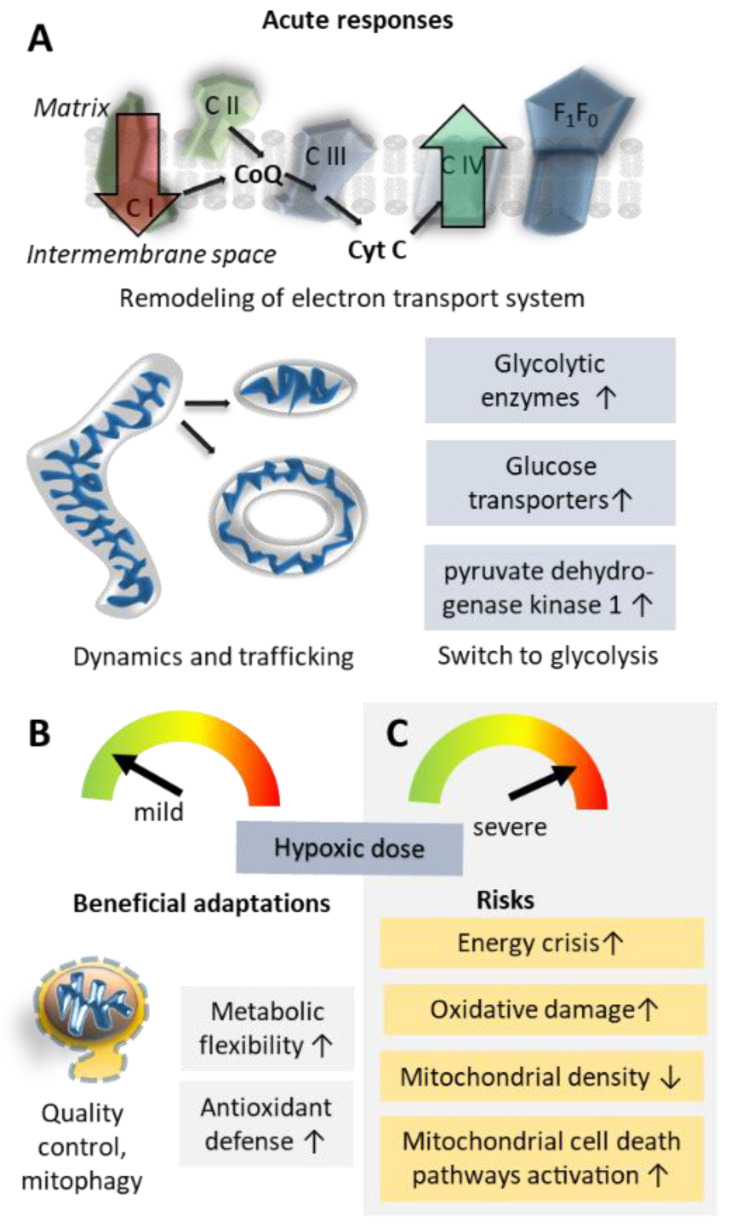
Mitochondrial responses to hypoxic stress. Acute responses of mitochondria to hypoxia (**A**) that lead to beneficial (**B**) or detrimental (**C**) outcomes, depending on the hypoxic dose and cellular resilience. CI—CIV, mitochondrial respiratory complexes I—IV; CoQ, coenzyme Q; Cyt C; cytochrome C; F_1_F_0_, F_1_F_0_ ATP synthase.

**Figure 4 ijms-24-01698-f004:**
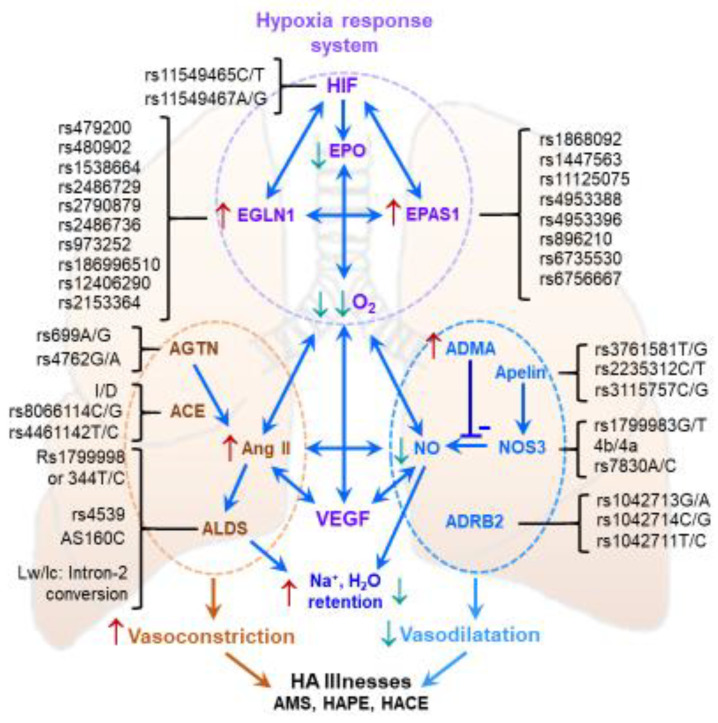
Genetic influence of the two major pathways on the physiological function.

**Table 1 ijms-24-01698-t001:** Gene polymorphisms impacting physiological and pathological responses to hypobaric hypoxia.

Oxygen-Sensing System		
Gene	Single-Nucleotide Polymorphisms	Physiological Function
*HIF-1α*	rs11549465C/T, rs11549467A/G	Regulates oxygen transport and delivery, glycolysis and many more
*EPAS1*	rs1868092A/G, 1447563C/A, rs11125075G/A, rs4953388A/G, rs4953396A/C, rs896210G/A, rs6735530C/T, rs6756667A/G	Expresses HIF-2α
*EGLN1*	rs1538664G/A, rs479200C/T, rs2486729G/A, rs2790879T/G, rs480902T/C, rs2486736G/A, rs973252A/G, rs186996510C/G, rs12406290A/G, rs2153364A/G	Regulates HIF-1α by promoting its hydroxylation and degradation
*EPO*	rs1617640A/C	Activates erythrocyte production
**Endothelial system**		
*AGT*	Rs699A/G, rs4762G/A	Angiotensinogen yields the precursor pentapeptide, a substrate for angiotensin converting enzyme (ACE)
*ACE*	I/D, 8066114C/G rs4461142T/C	ACE produces angiotensin II, a potent vasoconstrictor and activator of several signaling molecules, e.g., VEGF
*CYP11B2*	rs1799998 or –344T/C rs4539 5160C Iw/Ic: Intron-2 conversion	Mediates production of aldosterone, which activates sodium retention in the alimentary tract and kidneys to expand the extracellular fluid volume
*AGTR1*	Rs275651T/A, 275652T/G	Expresses a receptor that plays a major role in blood pressure homeostasis by governing angiotensin II signaling
*EDN1*	rs10478694 or -3A/-4A rs2070699G/T, rs5370	This gene produces a preproprotein that is cleaved to endothelin-1, a powerful vasoconstrictor
*APLN*	rs3761581T/G, rs2235312C/T, rs3115757C/G	Apelin stimulates NOS3 to produce NO
*NOS3*	rs1799983G/T, 4b/4a, rs7830A/C	The enzyme produces NO, a powerful vasodilator
*ADRB2*	rs1042713G/A rs1042714C/G rs1042711T/C	The protein stimulates vasodilatation. Several exonic SNPs have been reported
*CYBA*	rs9932581A/G or −930A/G rs4673C/T or H72Y	Mitochondrial respiratory complex component involved in ROS generation
*GSTP1*	rs1695A/G or I105V rs1138272C/T or A114V	The enzyme scavenges ROS to maintain homeostasis
**Vascular smooth muscle system**		
*VEGF*	rs3025039C/T	A factor promoting angiogenesis

## Data Availability

Not applicable.

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
