# Peer review of "Molecular Mechanisms of High-Altitude Acclimatization"

_ijms, 2023, doi:10.3390/ijms24021698_

Round 1

Reviewer 1 Report

The authors in this review manuscript summarized systemic physiological response to acute high-altitude exposure and pathophysiological development of HAIs including acute mountain sickness(AMS), high-altitude cerebral edema(HACE) and high-altitude pulmonary edema(HAPE). Next, the authors discussed underlying molecular mechanisms of pathogenesis of high-altitude hypoxia and associated acclimatization, including contribution of ROS  and mitochondria, role of transcription factors HIFs and Nrf2 as well as systemic energy and REDOX homeostatic, influence of individual genetic variation and time course of acclimatization. Last, based on the physiological responses and molecular mechanisms, the authors discussed practical strategies to help diagnosis or prediction of HAIs and to help preparation for high altitude exposure by targeting adaptive or maladaptive signaling pathways and molecular processes.

This review manuscript well covers the topic, summarizing systemic physiological response, pathogenesis and underlying mechanisms of high-altitude hypoxia and adaptation. The gap in knowledge is identified and the references are appropriately cited. The review is clear, comprehensive and well organized. The statements and conclusions drawn are supported by the listed citations. The figures/tables in the manuscript properly show and summarize the data cited in the manuscript, letting the data be easy to interpret and understand.

The authors mentioned the role of inflammation in HAIs, but didn’t further discuss it. Can the authors include data about the contribution of inflammation in the pathogenesis of HAIs, especially HAPE?

Author Response

Response to Reviewer 1

Thank you very much for the favorable evaluation and your valuable suggestion to discuss the potential role of inflammation in HAPE pathogenesis. Accordingly, we have added the following text [lines 202-207] and have cited two authoritative references:

While in high-altitude pulmonary hypertension (HAPH) inflammatory pathways may importantly contribute to the proliferation of pulmonary artery smooth muscle cells and pulmonary hypertension [1], this is likely not the case for HAPE. Although inflammation in HAPE could contribute to increased alveolar-capillary permeability, studies in humans rather indicate that inflammation constitutes a secondary response to the pulmonary edema and/or disruption of the alveolar-capillary barrier and not a primary factor in HAPE pathogenesis [2].”

  1. El Alam, S.; Pena, E.; Aguilera, D.; Siques, P.; Brito, J. Inflammation in Pulmonary Hypertension and Edema Induced by Hypobaric Hypoxia Exposure. Int J Mol Sci 2022, 23, doi:10.3390/ijms232012656.
  2. Swenson, E.R.; Bärtsch, P. High-altitude pulmonary edema. Compr Physiol 2012, 2, 2753-2773, doi:10.1002/cphy.c100029.

Reviewer 2 Report

NA

This is good piece.

Most of these contents are out there over and over again by the authors (either individually or with their group). So basically, I find it re-hashed and repackaged collaboratively.

Some of these authors have recently published about ROS and hypoxia with similar contents.

One aspect, authors need to avoid and demarcate is that they should not mix acclimatization with adaptation which they have done here. If you are talking about acute exposure and acclimatization, focus there. Or, if you want to do all of them together, organize properly. 

Anything regarding adaptation should be removed from this review.

Author Response

Response to Reviewer 2

Thank you very much for the favorable evaluation and valuable suggestions. We agree, several parts of the present review have been repeatedly discussed in the literature. However, in this manuscript we focused on merging knowledge from different fields, e.g., physiology, mitochondrial and molecular biology, and genetics, to provide a comprehensive review of acclimatization to high altitude, the underlying mitochondrial and molecular mechanisms, and the potential impact of gene polymorphisms on high altitude acclimatization in humans.

As the reviewer recommends, we have removed all discussion of adaptation from the manuscript in order to avoid confusion. The term “adaptation” now appears only once, in reference to “genetic adaptation” in highlanders [line 138 of manuscript text file].

Reviewer 3 Report

Please address the following question: The authors nicely indicated that high altitude, in addition to hypoxia as the main factor, is also characterized by other potentially extreme conditions, such as cold, increased solar radiation and lower humidity. Can you add at least a paragraph describing whether there are (or not) the potential influences of other ecological factors to the development of acute high altitude illnesses?

Author Response

Response to Reviewer 3

We are sincerely grateful for your favorable evaluation and your helpful and constructive comments. We agree, hypobaric hypoxia per se is by no means the only environmental factor imposed by high altitude. Accordingly, we have added the following paragraph [lines 211-223 of the manuscript text file] which briefly considers the contributions of hypothermia and of dehydration, alone or combined with heavy exercise, to the risk of high-altitude illnesses:

It must be mentioned that apart from hypoxia, several other environmental and/or behavioral factors could contribute to the heightened risk of serious illnesses at high-altitudes [1]. Although a comprehensive examination of these factors and of their interactions with hypoxia is beyond the scope of this review, the potential contributions of hypothermia and dehydration to HAIs merit discussion. For example, exposure to cold at high altitude may predispose mountaineers to dehydration due to elevated cold-diuresis and poor access to fluids [2]. These authors identified an association between the level of dehydration and the risk of AMS [2]. Moreover, exposure to cold and altitude may increase risk of thrombosis and myocardial infarction [3]. Heavy exercise at altitude exacerbates dehydration, raising the risk of rhabdomyolysis and acute liver injury [4], and intensifies arterial hypoxemia thereby triggering or accelerating the development of AMS [5]. Furthermore, both severe low ambient temperature and intense physical activity are also implicated as predisposing factors for HAPE development, primarily due to the increase in pulmonary artery pressure [6,7]”.

  1. Boggild, A.K.; Costiniuk, C.; Kain, K.C.; Pandey, P. Environmental hazards in Nepal: altitude illness, environmental exposures, injuries, and bites in travelers and expatriates. J Travel Med 2007, 14, 361-368, doi:10.1111/j.1708-8305.2007.00145.x.
  2. Nerín, M.A.; Palop, J.; Montaño, J.A.; Morandeira, J.R.; Vázquez, M. Acute mountain sickness: influence of fluid intake. Wilderness Environ Med 2006, 17, 215-220, doi:10.1580/1080-6032(2006)17[215:amsiof]2.0.co;2.
  3. Whayne, D.F. Altitude and cold weather: are they vascular risks? Curr Opin Cardiol 2014, 29, 396-402, doi:10.1097/HCO.0000000000000064.
  4. Yeh, Y.C.; Chen, C.C.; Lin, S.H. Case report: severe rhabdomyolysis and acute liver injury in a high-altitude mountain climber. Front Med 2022, 9, 917355, doi:10.3389/fmed.2022.917355.
  5. Roach, R.C.; Maes, D.; Sandoval, D.; Robergs, R.A.; Icenogle, M.; Hinghofer-Szalkay, H.; Lium, D.; Loeppky, J.A. Exercise exacerbates acute mountain sickness at simulated high altitude. J Appl Physiol (1985) 2000, 88, 581-585, doi:10.1152/jappl.2000.88.2.581.
  6. Schoene, R.B.; Roach, R.C.; Hackett, P.H.; Harrison, G.; Mills, W.J. High altitude pulmonary edema and exercise at 4,400 meters on mount McKinley. Effect of expiratory positive airway pressure. Chest 1985, 87, 330-333.
  7. Swenson, E.R. Early hours in the development of high-altitude pulmonary edema: time course and mechanisms. J Appl Physiol (1985) 2020, 128, 1539-1546, doi:10.1152/japplphysiol.00824.2019.